



# Molecular level characterization of supraglacial dissolved organic matter sources and exported pools the southern Greenland Ice Sheet

Eva L. Doting[1,2,*], Ian T. Stevens[1], Anne M. Kellerman[3], Pamela E. Rossel[4], Runa Antony[4,5], Amy M. McKenna[6, 7], Martyn Tranter[1], Liane G. Benning[4, 8], Robert G. M. Spencer[3], Jon R. Hawkings[2, 9] and Alexandre M. Anesio[1]

[1] Department of Environmental Science, iClimate, Aarhus University, Frederiksborgvej 399, 4000 Roskilde, Denmark
[2] Department of Earth and Environmental Science, University of Pennsylvania, Philadelphia, PA, USA
[3] National High Magnetic Field Laboratory Geochemistry Group and Department of Earth, Ocean, and Atmospheric Science, Florida State University, Tallahassee, FL, USA
[4] Interface Geochemistry Section, German Research Centre for Geosciences, GFZ Potsdam, Telegrafenberg, 14473 Potsdam, Germany
[5] National Centre for Polar and Ocean Research, Ministry of Earth Sciences, Goa, India
[6] National High Magnetic Field Laboratory, Florida State University, Tallahassee, Florida, 32310-4005, USA
[7] Department of Soil and Crop Sciences, Colorado State University, Fort Collins, CO, 80523, USA
[8] Department of Earth Science, Freie Universität Berlin, 12249 Berlin, Germany
[9] iC3, Department of Geosciences, UiT The Arctic University of Norway, Tromsø, Norway

*Correspondence to*: Eva L. Doting (edoting@sas.upenn.edu)

**Abstract.** During the ablation season, active microbial communities colonise large areas of the Greenland Ice Sheet surface and produce dissolved organic matter (DOM) that may be exported downstream by surface melt. Meltwater flow through the bare ice interfluvial area, characterized by a porous weathering crust, is slow ($\sim 10^{-2}$ m d$^{-1}$), meaning that it presents a potential site for photochemical and/or microbial alteration of supraglacial DOM. Transformations of supraglacial DOM during transport through the supraglacial drainage system remain unexplored, limiting our understanding of supraglacial DOM inputs to downstream subglacial and coastal ecosystems. Here, we employ negative-ion electrospray ionization 21 tesla Fourier transform ion cyclotron resonance mass spectrometry to catalogue the molecular composition of DOM in supraglacial dark ice, weathering crust meltwater, and supraglacial stream water sampled in a hydrologically connected supraglacial micro-catchment to address this knowledge gap. Dark ice DOM contained significantly more aromatic ($25 \pm 3$ %) and less biolabile ($13 \pm 4$ %) DOM than weathering crust meltwater ($3 \pm 0$ and $50 \pm 0$ %, respectively), pointing to retention of DOM on the ice surface and microbial, as well as photochemical alteration of DOM during transit through the supraglacial drainage system. These findings have implications for our understanding of supraglacial biogeochemical cycling, highlighting the importance of including the weathering crust photic zone when assessing supraglacial inputs to subglacial and downstream ecosystems.



## 1 Introduction

Microbial blooms dominated by the algae *Ancylonema alaskanum* and *Ancylonema nordenskiöldii* (Procházková et al., 2021; Lutz et al., 2018) cover large areas of the Greenland Ice Sheet ablation zone during the summer melt season (Cook et al., 2020;
Uetake et al., 2010; Stibal et al., 2012). These algae produce a purple-brown pigment, purpurogallin carboxylic acid-6-O-β-D-glucopyranoside (Remias et al., 2012), which provides protection from UV radiation and significantly lowers the albedo of the Greenland Ice Sheet ablation zone (Uetake et al., 2010; Yallop et al., 2012; Stibal et al., 2017; Ryan, 2017, 2018). Ice surface microbial production was found to correlate with supraglacial concentrations of carbohydrates and low-molecular weight compounds (Musilova et al., 2017), indicating that microbial communities are likely the primary driver of biolabile dissolved
organic matter (DOM) production on the ice surface. Previous studies have shown that supraglacial stream and supraglacial snowpack DOM contain a high relative abundance of biolabile (D'Andrilli et al., 2015) aliphatic and peptide-like molecular formulae (Stubbins et al., 2012; Kellerman et al., 2021; Hemingway et al., 2019; Antony et al., 2017; Lawson et al., 2014), which are likely of microbial origin (Kellerman et al., 2018; Spencer et al., 2015). Yet, to date, DOM associated with algal blooms on glacier surface ice has not been characterized at the molecular level, limiting our understanding of surface ice
microbial contributions to the biolabile character of supraglacial stream DOM.

The biolability of glacial runoff has been shown to correlate with increasing $^{14}$C age, meaning that glaciers are a source of both ancient and biolabile DOM (Hood et al., 2009; Stubbins et al., 2012; Spencer et al., 2014b; Singer et al., 2012). Radiocarbon dating of DOC in glacier ice and meltwaters from Alaska, the European Alps, Greenland, the Tibetan Plateau, and Ecuador all
confirmed the presence of ancient carbon in glacial runoff (Bhatia et al., 2013; Stubbins et al., 2012; Singer et al., 2012; Spencer et al., 2014a, b; Holt et al., 2023). Aged DOM may be delivered to supraglacial surfaces by atmospheric deposition of soil or combustion-derived organic matter (Hood et al., 2009; Barker et al., 2009; Bhatia et al., 2010; Singer et al., 2012; Fellman et al., 2015; Li et al., 2018; Price et al., 2009; Spencer et al., 2014b; Stubbins et al., 2012; Bardgett et al., 2007; Holt et al., 2023). However, this source material would be expected to be characterized by high aromaticity (Chen and Jaffé, 2014;
Fellman et al., 2013; Hansen et al., 2016; Li et al., 2018; Masiello, 2004) and hence appears disconnected from the aliphatic and peptide-rich DOM observed in supraglacial runoff. However, Holt et al. (2021) showed that photochemical degradation of both modern and aged aromatic organic matter sources common to glacier environments produces aliphatic compounds, potentially explaining the composition of glacial DOM. It is not yet understood whether this potential photochemical alteration of allochthonous DOM occurs during transport to glacier surfaces, on the glacier surface itself, or both.

To assess potential transformations of allochthonous and autochthonous DOM, the transport of water and associated DOM through the supraglacial drainage network must be considered. On the Greenland Ice Sheet, the supraglacial drainage network is comprised of streams, rivers, lakes and more than 200,000 km$^2$ of bare ice surface (Ryan et al., 2019). The majority of surface melt is generated in this bare ice interfluvial area (Steger et al., 2017), which is characterised by the presence of a

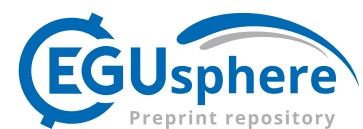

porous weathering crust, which forms due to shortwave radiation penetration into the surface ice, the consequent near-surface melting that arises, and the rate of percolation away from the melt zone (Stevens et al., 2018). Interstitial waterflow is slow (~ $10^{-2}$ m d$^{-1}$), meaning that this weathering crust photic zone is a likely site for microbial and/or photochemical alteration of supraglacial DOM because the water residence time here is days to weeks (Irvine-Fynn et al., 2021; Stevens et al., 2018; Yang et al., 2018). Typically, surface melt ends up in supraglacial streams that flow into moulins and crevasses that drain to the

glacier bed (Yang and Smith, 2016). An understanding of supraglacial DOM characteristics and sources, and its transformations during transport through the supraglacial drainage system is therefore critical to assessing the nature and biolability of supraglacial DOM inputs to downstream subglacial and coastal ecosystems.

Here, we assess changes in DOM composition as it is transported through the supraglacial drainage system on the Greenland

Ice Sheet. We use negative electrospray ionization 21 Tesla Fourier transform ion cyclotron resonance mass spectrometry (21 T FT-ICR MS) to determine the molecular composition of dark surface ice, laboratory-generated dark surface ice debris leachate, weathering crust meltwater, and supraglacial stream meltwater sampled from a hydrologically connected micro-catchment. We reveal significant differences between supraglacial DOM pools, demonstrating the importance of the weathering crust in supraglacial biogeochemical processes with respect to the composition of supraglacial DOM that is

delivered to downstream subglacial and coastal ecosystems.

## 2 Methods

### 2.1 Site description, sample collection and field processing

A small supraglacial catchment on the southern Greenland Ice Sheet (61° 06' N, 46° 51' W; Fig. 1), located < 1 km from the QAS_M PROMICE weather station (Fausto et al., 2021), was sampled on July 28, 2021 (Day of Year (DOY) 209). Within

the catchment, samples were collected from surface ice, weathering crust meltwater (by sampling a refilled auger hole), and supraglacial stream water (Fig. 1B-E). These were supplemented with measurements of hydraulic conductivity and discharge in weathering crust auger holes A-E (Fig. 1B) and the supraglacial stream, respectively.

Samples for FT-ICR MS molecular level characterization were collected from surface ice between 14:00 and 15:00 (n = 4),

weathering crust meltwater from auger hole D at 14:00 (n = 4), and supraglacial stream water at Q (n = 5) in Fig. 1. Surface ice samples (herein "dark ice"; Fig 1C) were collected from areas with visible debris and algae (observed by hand-held microscope in the field), using a sample-cleaned ice axe to scrape the top ~2 cm of ice into acid cleaned (1.2 M HCl) 1 L polycarbonate bottles. Ice was melted, filtered (combusted 47 mm, 0.7 μm GF/F filter, Whatman), acidified (pH 2, HCl). Aliquots for DOC analysis were stored in furnaced 40 mL amber vials with acid-washed caps and PTFE-lined septa in the

dark at 4 °C, while the rest of the acidified sample was stored in the dark in an acid cleaned PC bottle until solid phase extraction back in the home laboratory. Material retained on the filter (herein "surface debris") was collected into 150 mL acid-cleaned



polycarbonate bottles and stored in the dark at -20 °C. Weathering crust meltwater and supraglacial stream water samples were collected into 1 L PC bottles and processed as per the surface ice filtrate.

**Figure 1: (A) Map of Greenland indicating the approximate location of the study site; (B) drone image of the supraglacial catchment indicating water flow direction and sampling locations (indicated by Q for stream, A-E for weathering crust holes and SI for approximate area where dark surface ice was sampled; (C) close-up of typical dark surface ice, (D) close-up of typical weathering crust auger hole (diameter 14 cm); (E) close-up of the stream sampling location; and (F) schematic illustrating the sampling area with water flow direction, a weathering crust auger hole and the supraglacial stream.**

## 2.2 Near-surface hydrology

Recharge rate was measured in weathering crust auger holes at odd hours between 7:00 and 21:00, with a supplementary measurement at 14:00, using logging ultrasonic range finders. Hydraulic conductivity and water table height were calculated following Stevens et al. (2018). Combined with an uncrewed aerial vehicle (UAV)-derived orthophoto and digital elevation model (DEM), weathering crust meltwater flow direction and magnitude were modelled using the Spatial Analyst package in



ArcMap 10.8 (esri, USA), designed for terrestrial groundwater systems. A full description is included in the supplementary methods.

## 2.3 Preparation of laboratory leachates

Within four months of sample collection, surface debris samples, which had been kept at -20 °C in the dark, were freeze-dried
(ScanVac CoolSafe, -110°C) and cryo-milled using stainless-steel sample holders (Retsch MM400). Sample holders were cleaned with Milli-Q and the first aliquot of milled sample was discarded. For each sample, 100 mg of milled material was weighed into acid-cleaned and combusted 4 mL glass vials, in triplicate, and 1 mL of Milli-Q was added to each triplicate to extract DOM that may feasibly be leached from debris on the ice surface. Vials were shaken at 500 rpm for 1 hour and then centrifuged at 3200 rpm for 10 minutes. To yield enough volume for DOC and FT-ICR MS analysis, supernatants were diluted
in 50 mL Milli-Q water and filtered using an acid-washed and combusted glass syringe and pre-rinsed wwPTFE (Acrodisc One, 0.2 μm, Pall) syringe filter. A sub-sample for DOC analysis was collected from each prepared sample before recombining the triplicates for solid phase extraction (Section 2.6). The laboratory-generated cold water surface ice debris leachate is referred to as 'laboratory leachate' throughout.

## 2.4 Dissolved organic carbon analysis

DOC measurements were carried out on a Shimadzu TOC-L$_{CSH}$ analyser with a high sensitivity platinum catalyst within four months of sample collection (samples were stored at 4 °C in the dark between collection and analysis). Samples were analysed in the non-purgeable organic carbon mode. Pre-acidified samples were sparged with carbon-free air for 2 min to eliminate inorganic carbon species before oxidizing the remaining DOC to $CO_2$ through high-temperature combustion (680 °C), followed by non-dispersive infrared detection. Up to five replicate injections were made for each sample until the coefficient of variation
(CV) for three of the replicate injections was ≤ 2 %. Measurements were quantified using a potassium hydrogen phthalate (Sigma-Aldrich) calibration curve. The instrument quantification limit (27 μg L$^{-1}$) was calculated from linear calibrations following the root mean square error method described by Corley (2003). Analytical precision calculated based on the standard error from seven repeat measurements of a 100 μg L$^{-1}$ standard was 1.6 %.

## 2.5 Total carbon and total nitrogen analysis

Cryo-milled surface debris samples were analysed for total carbon (TC) and total nitrogen (TN) content on an Elemental analyser (Euro EA). Samples were oxidized in an oxygen atmosphere at a furnace temperature of 950 °C. After combustion, the resulting gases ($CO_2$, $NO_x$) were separated in a gas chromatography column at 70 °C and detected by thermal conductivity. The limit of quantification was 0.15 % for both nitrogen and carbon. Overall precision for analyses of carbon and nitrogen was within 5 % RSD. Results are reported in weight percent (wt. %).



## 2.6 Solid phase extraction

All DOM samples were solid phase extracted back in the home laboratory, following Dittmar et al. (2008), to remove inorganic interferences and concentrate the organic matter prior to FT-ICR MS analysis. Laboratory leachates were extracted on 3 mL, 100 mg, Bond Elut PPL cartridges. Weathering crust water, stream water and dark ice samples were extracted on 6 mL, 1 g, Bond Elut PPL cartridges. Samples were eluted with 6 mL of methanol into acid-soaked and combusted 10 mL amber glass vials. Eluates were dried under nitrogen flow and stored at -20 °C until analysis.

## 2.7 21 T Fourier Transform Ion Cyclotron Resonance Mass Spectrometry

Dried eluates were reconstituted in methanol prior to analysis, adjusting the volume to achieve a target concentration of 50 mg C L$^{-1}$. DOM composition was analysed using a custom-built hybrid linear ion trap FT-ICR MS equipped with a 21 tesla superconducting solenoid magnet at the National High Magnetic Field Laboratory in Tallahassee, Florida (see supplementary methods; Hendrickson et al., 2015; Smith et al., 2018). Time-domain transients of 3.1 s (achieved mass resolving power, $m/\Delta m_{50\%} > 2,000,000$ at $m/z$ 400) were conditionally co-added and acquired with the Predator data station, with 100 time-domain acquisitions averaged for all experiments (Blakney et al., 2011), phase-corrected (Xian et al., 2010), and internally-calibrated with 10-15 homologous series that span the entire molecular weight distribution based on the "walking" calibration method (Savory et al., 2011).

Mass spectra were calibrated, and molecular formulae were assigned ($C_{1-100}H_{4-200}O_{1-30}N_{0-4}S_{0-2}$). Molecular formulae were classified by heteroatom (any atom other than carbon or hydrogen) type and number as oxygen only (CHO), oxygen and nitrogen (CHON), oxygen and sulphur (CHOS), and oxygen, nitrogen, and sulphur (CHONS). Molecular speciation was categorized based on neutral elemental ratios of H/C and O/C and the modified aromaticity index (AI$_{mod}$), calculated from the neutral species following Eq. (1):

$$AI_{mod} = \frac{1 + C - \frac{1}{2}O - S - \frac{1}{2}(N+H)}{C - \frac{1}{2}O - N - S} \tag{1}$$

where C, H, O, N and S are the number of carbon, hydrogen, oxygen, nitrogen and sulphur atoms in a given molecular formulae (Koch and Dittmar, 2006, 2016). The nominal oxidation state of carbon (NOSC) was calculated following Eq. (2):

$$NOSC = 4 - \frac{(4C + H - 3N - 2O - 2S)}{C} \tag{2}$$

Where C, H, O, N and S denote the number of atoms of each element in each formula (Riedel et al., 2012). Elemental compositions were classified into eight groups: condensed aromatics (AI$_{mod} \geq 0.67$), polyphenols ($0.67 > AI_{mod} > 0.50$), peptide-like formulae (H/C $\geq 1.5$, O/C $\leq 0.9$, N > 0), sugar-like formulae (H/C $\geq 1.5$, O/C > 0.9), and highly unsaturated and phenolic formulae (HUP; AI$_{mod} \leq 0.50$, H/C < 1.5) and aliphatics (H/C $\geq 1.5$, O/C $\leq 0.9$, N = 0), which were both separated





into high O/C (O/C > 0.5) and low O/C (O/C < 0.5) (Osterholz et al., 2016; Spencer et al., 2014b). The relative abundance (RA) of each assigned formula in a sample was obtained by dividing the signal magnitude of each individual $m/z$ peak by the sum of all assigned signals in the sample. RA weighted metrics were calculated for the mass, $AI_{mod}$, NOSC, H/C and O/C. Calculated metrics can give a variety of information about the composition and lability of DOM in studied samples. For example, compounds with a H/C > 1.5 have been related to more biolabile material, whereas compounds with a H/C < 1.5 are considered less biolabile (D'Andrilli et al., 2015). A negative NOSC corresponds to more reduced compounds, whereas a positive NOSC corresponds to more oxidized ones. It has to be noted that these boundaries can be ambiguous as, for example, the glacier ice algae polyphenolic pigment purpurogallin carboxylic acid-6-O-β-D-glucopyranoside ($C_{18}H_{18}O_{12}$) contains a two-ring aromatic core (Remias et al., 2012), but does not meet the $AI_{mod}$ threshold for polyphenolics due to its glucopyranoside sidechain and is hence classified as a high O/C HUP. However, the formulae classifications outlined above have shown systematic and biogeochemical coherence with sources and degradation patterns of DOM (Stubbins et al., 2010; D'Andrilli et al., 2015; Kellerman et al., 2018; Spencer et al., 2015) and are therefore informative for use in the present study.

**2.8 Statistics**

All statistical analyses were performed in R (R Team, 2014). Peaks with a high intensity (relative abundance > 0.1%) in the procedural field blank were removed from the dataset as they were assumed to be contaminants, and data were renormalized to the total sum of assigned signals before exploring DOM characteristics. Pairwise comparisons were performed between samples grouped as laboratory leachate, dark ice, weathering crust meltwater, and supraglacial stream water. Bartlett's test was used to assess homogeneity of variance for all variables. If variance was equal, a one-way ANOVA assuming equal variance was used, followed by pairwise comparison using a t-test if the ANOVA was significant, and p-values were adjusted using Bonferroni correction. If variance was unequal, a one-way ANOVA assuming unequal variance was used, followed by pairwise comparison using a t-test assuming unequal variance if the ANOVA was significant, and p-values were adjusted using Šidák correction. Prior to Principal Component Analysis (PCA) using the R package 'vegan' (Oksanen et al., 2011), variables were unit variance scaled.

**3 Results**

**3.1 Near-surface hydrology of the study site**

Our hydrological modelling approach reveals that meltwater from Hole D, where weathering crust meltwater samples were collected, transits from the sampling point within the weathering crust in a south-easterly direction to the main supraglacial stream over a period of nine days (Figure 2) assuming the prevailing weather crust state during the study period remains constant.



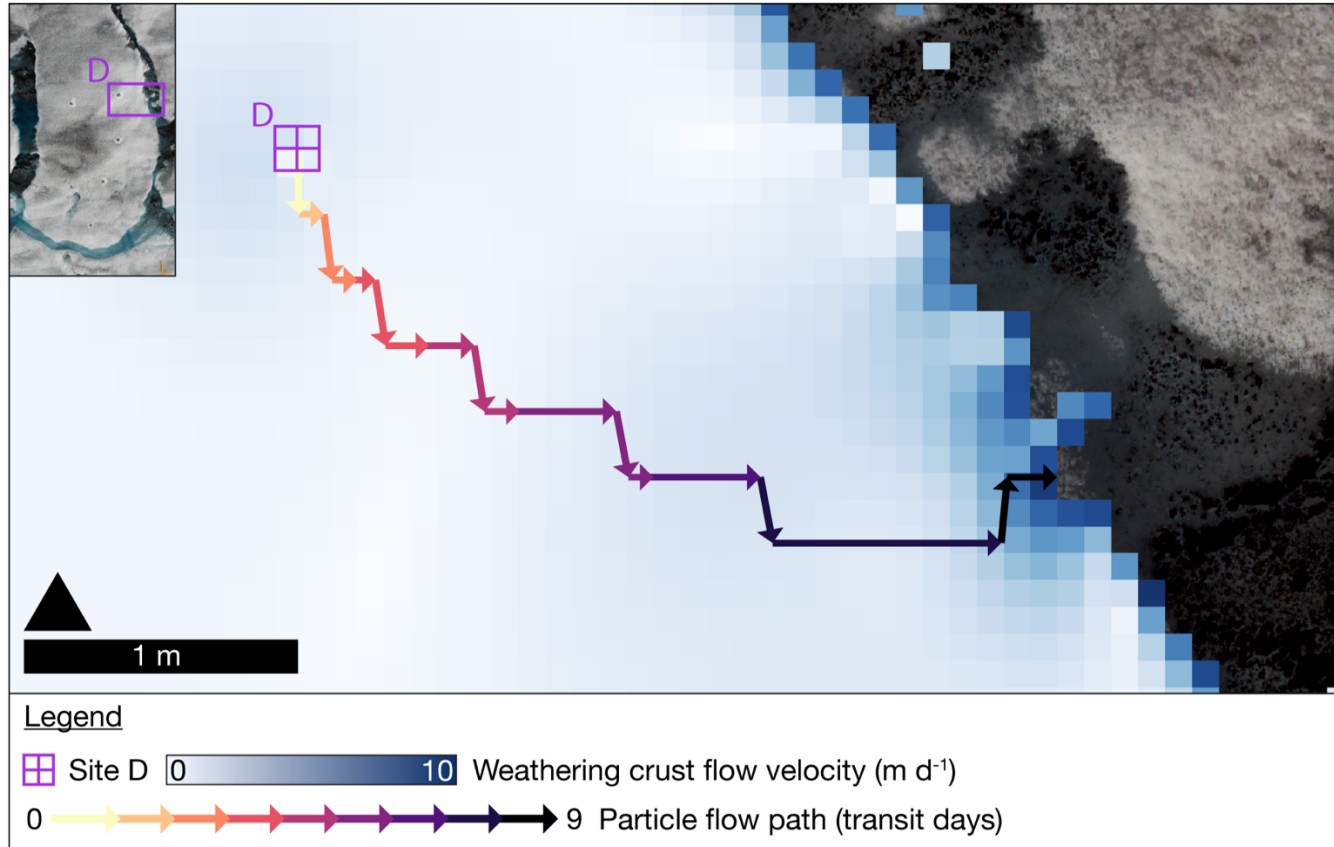

**Figure 2: Results of the flow direction and magnitude model, with pixel colour indicating weathering crust water flow velocity. Arrows indicate the modelled particle flow path through the weathering crust from Hole D to the supraglacial stream.**

**3.2 Bulk DOC, TC and TN concentrations**

Dark ice DOC concentrations were similar (0.90 - 1.01 mg L$^{-1}$) in the four samples collected within the micro-catchment and were significantly higher than the DOC concentrations in weathering crust meltwater and supraglacial stream water (Table 1). The surface ice debris, from which the laboratory leachate was generated, contained between 3.00 – 3.35 wt. % TC and between 0.25 – 0.28 wt. % TN. The mean coefficient of variation of DOC concentrations between triplicate extractions of surface debris of the same sample was 2.4%.

**3.3 Molecular level composition of supraglacial DOM pools**

A total of 24,578 unique molecular formulae were assigned across the dataset, with between 6,385 and 9,667 formulae assigned in individual samples (Table 1). There was no significant difference between sample types (laboratory leachate, dark ice, weathering crust and supraglacial stream water) in terms of the number of formulae assigned, and 2,885 formulae were shared between all samples in the dataset. For laboratory leachate samples, on average 63% (3,653 – 6,053 formulae) of the assigned formulae were also present in the corresponding dark ice sample. The mean RA weighted mass (Table 1) was higher in dark



ice (450 ± 10 Da) and supraglacial stream water (452 ± 12 Da) samples than in the laboratory leachates (407 ± 10 Da). Mean RA weighted NOSC and H/C ratio were significantly lower and higher, respectively, in laboratory leachate, weathering crust meltwater and supraglacial stream than in dark ice (Table 1), corresponding to a significantly lower prevalence of aliphatic

and peptide-like compounds (13 ± 4 %RA) in dark ice DOM than in the rest of the dataset (32.8 – 68.3 % RA).

The %RA of heteroatom classes differed significantly between sample types (Table 1). Laboratory leachates were composed of approximately equal portions of CHO (35.9 – 58.0 %RA) and CHOS (33.3 – 55.2 %RA), with relatively minor contributions of CHON (4.9 – 10.6 %RA) and CHONS (0.25 – 0.47 %RA) formulae. Dark ice DOM was predominantly composed of CHO

(91.2 – 96.5 %RA), with minor contributions of CHON (3.5 – 6.9 %RA) and CHOS (0 – 1.9 %RA) formulae. The prevalence of CHON in weathering crust meltwater and supraglacial stream samples was similar (13.9 – 15.5 and 14.9 – 16.7 %RA, respectively), but there was a significant difference in the contribution of CHO (63.3 – 70.5 and 73.6 – 83.8 %RA, respectively) and CHOS (14.5 – 22.1 and 0.6 – 11.5 %RA, respectively). No CHONS formulae were assigned in dark ice, weathering crust meltwater, or supraglacial stream samples.


Laboratory leachate, weathering crust meltwater, and supraglacial stream DOM was predominantly composed of aliphatic and HUP compounds (Table 1). Aliphatic compounds accounted for approximately half of the DOM in laboratory leachates (46.4 – 62.1 %RA), just under half in weathering crust meltwater samples (40.8 – 47.4 %RA) and roughly a third of DOM in supraglacial stream samples (28.4 – 38.3 %RA). Dark ice DOM was comprised predominantly of HUP (59.0 – 68.5 %RA)

and aromatic (21.8 – 27.5 %RA) compounds. High O/C HUP compounds with the molecular formula $C_{18}H_{18}O_{12}$, which may include the algal pigment purpurogallin carboxylic acid-6-O-β-D-glucopyranoside, accounted for 7.3 – 13.9 %RA of dark ice DOM, 2.1 – 7.4 %RA of laboratory leachate DOM, 0.02 – 0.04 %RA of weathering crust meltwater DOM, and 0.01 %RA of supraglacial stream DOM. Polyphenolic compounds with the molecular formula $C_{12}H_8O_7$, which may include the aglycone degradation product of the algal purpurogallin pigment, accounted for 5.7 – 8.8 %RA of dark ice DOM, 0.2 – 2.5 %RA of

laboratory leachate DOM, 0.02 %RA of weathering crust meltwater DOM, and 0.01 %RA of supraglacial stream DOM. Aromaticity in dark ice was high ($AI_{mod}$ 0.305 – 0.340) compared to laboratory leachate, weathering crust meltwater, and supraglacial stream samples ($AI_{mod}$ 0.123 – 0.172), in which aromatic compounds made up less than 5 %RA of DOM.

The number of formulae, or molecular diversity, of aromatic compounds assigned was highest for dark ice (765 – 957),

followed by weathering crust meltwater (491 – 547), supraglacial stream water (271 – 420), and was lowest in laboratory leachates (153 – 440). The number of biolabile (aliphatic and peptide-like) formulae was highest in laboratory leachate and weathering crust samples (2,984 – 3,719 and 3,124 – 3,341, respectively), and lowest in dark ice (1,584 – 2,640). The number of biolabile formulae assigned in supraglacial stream samples (2,498 – 2,890) was similar to weathering crust meltwater samples. The %RA of peptide-like compounds was significantly lower in dark ice (1.3 – 2.8 %RA) than in other sample types

(Table 1).



**Table 1: Dissolved organic carbon concentrations and dissolved organic matter composition for each sample type, expressed as mean (standard deviation). %RA denotes precent relative abundance, # denotes number of formulae. Significance (adjusted p-value < 0.05) of pairwise comparisons is denoted by capital letters, where values that have at least one letter in common are not significantly different from one another.**

| | Laboratory leachate | Dark ice | Weathering crust | Stream |
|---|---|---|---|---|
| **DOC (mg L$^{-1}$)** | 386 (85) [A*] | 0.94 (0.05) [B] | 0.18 (0.04) [C] | 0.14 (0.01) [C] |
| **Formulae (#)** | 8,403 (864) [A] | 7,570 (965) [A] | 9,008 (587) [A] | 8,343 (1022) [A] |
| **Mass$^{wa}$ (Da)** | 407 (10) [A] | 450 (10) [BC] | 429 (9) [AB] | 452 (12) [C] |
| **AI$_{mod}$$^{wa}$** | 0.149 (0.022) [A] | 0.326 (0.016) [B] | 0.158 (0.003) [A] | 0.162 (0.006) [A] |
| **NOSC$^{wa}$** | -0.652 (0.120) [A] | 0.112 (0.061) [B] | -0.569 (0.045) [AC] | -0.470 (0.043) [C] |
| **H/C$^{wa}$** | 1.471 (0.056) [A] | 1.072 (0.033) [B] | 1.438 (0.013) [A] | 1.405 (0.019) [A] |
| **O/C$^{wa}$** | 0.369 (0.039) [A] | 0.582 (0.019) [B] | 0.406 (0.018) [AC] | 0.445 (0.014) [C] |
| **CHO (%RA)** | 47 (9) [A] | 94 (2) [B] | 68 (3) [C] | 81 (4) [D] |
| **CHON (%RA)** | 8 (2) [A] | 5 (2) [A] | 15 (1) [B] | 16 (1) [B] |
| **CHOS (%RA)** | 44 (9) [A] | 1 (1) [B] | 17 (3) [C] | 3 (5) [B] |
| **CHONS (%RA)** | 0.37 (0.09) [A] | 0 (0) [B] | 0 (0) [B] | 0 (0) [B] |
| **Aliphatic High O/C (%RA)** | 5 (2) [AB] | 4 (1) [B] | 7 (1) [AC] | 9 (1) [C] |
| **Aliphatic Low O/C (%RA)** | 51 (7) [A] | 7 (2) [B] | 37 (3) [C] | 25 (4) [D] |
| **HUP High O/C (%RA)** | 21 (8) [A] | 50 (7) [A] | 21 (3) [A] | 25 (3) [A] |
| **HUP Low O/C (%RA)** | 15 (2) [AB] | 12 (3) [B] | 26 (1) [A] | 34 (2) [A] |
| **Peptide-like (%RA)** | 5 (2) [A] | 2 (1) [B] | 7 (1) [A] | 5 (1) [A] |
| **Condensed aromatic (%RA)** | 0 (0) [AC] | 8 (1) [B] | 1 (0) [C] | 0 (0) [A] |
| **Polyphenolic (%RA)** | 2 (2) [A] | 17 (2) [B] | 3 (0) [CD] | 2 (0) [AD] |
| **Sugar (%RA)** | 0.17 (0.10) [A] | 0.03 (0.02) [B] | 0.03 (0.02) [B] | 0.04 (0.02) [B] |
| **Aromatic compounds (%RA)** | 3 (2) [AB] | 25 (3) [C] | 3 (0) [A] | 2 (0) [B] |
| **Aliphatic + peptide-like (%RA)** | 62 (8) [A] | 13 (4) [B] | 50 (2) [C] | 38 (4) [D] |
| **Aromatic compounds (#)** | 286 (134) [A] | 862 (80) [B] | 523 (24) [C] | 362 (70) [AC] |
| **Aliphatic + peptide-like (#)** | 3,395 (320) [A] | 2,088 (464) [B] | 3,228 (119) [AC] | 2,698 (148) [C] |

[*] DOC concentration in supernatant (1 mL) after cold water extraction of 100 mg cryo-milled dark surface ice debris

[wa] Relative abundance weighted average





### 3.3 Compositional differences between hydrologically connected DOM pools

Principal component (PC) analysis was used to assess DOM parameters that distinguish the different sample groups in more
detail (Fig. 3). PC1 explained 63% of variance in the data and correlated positively with NOSC, AI$_{mod}$, high O/C HUP,
condensed aromatic and polyphenolic compounds, as well as %RA of CHO, and mean RA weighed mass. PC1 correlated
negatively with molecular diversity (number of formulae), peptide-like compounds, low and high O/C aliphatic compounds,
low O/C HUP compounds, and %RA of CHON and CHOS. Dark ice samples separated from the other sample groups along

PC1, reflecting the higher aromaticity and lower relative abundance of biolabile peptide-like and aliphatic compounds. PC2
explained a further 26% of variance in the data, correlating positively with low O/C HUP and high O/C aliphatic compounds,
%RA of CHO, %RA of CHON, and mean RA weighted mass. Finally, PC2 correlated negatively with low O/C aliphatic
compounds and %RA CHOS. Laboratory leachate, weathering crust meltwater and supraglacial stream samples separate along
PC2, with the former containing a higher %RA of CHOS formulae and low OC aliphatic compounds. Weathering crust

meltwater and supraglacial stream samples formed two separate clusters, driven by differences in prevalence of low O/C
aliphatic compounds and sulphur-containing compounds. Overall, the four sample types present significantly different DOM
compositions.

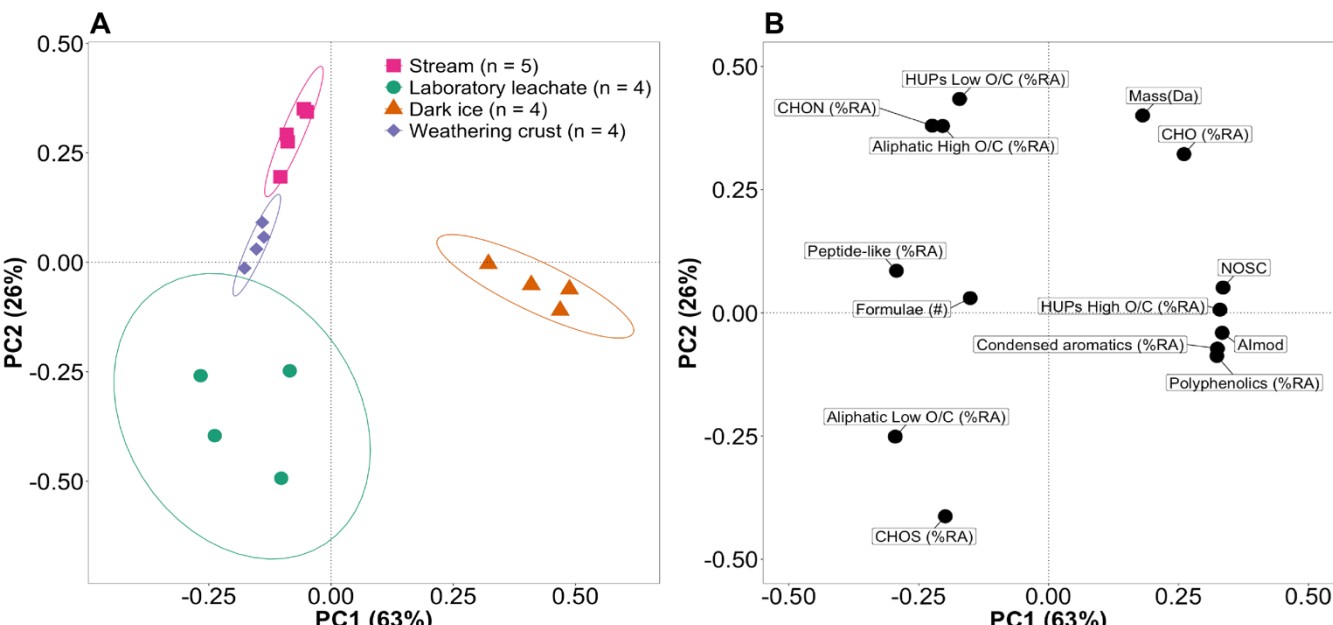

**Figure 3: (A) PC analysis scores plot of DOM composition in laboratory leachate (green circles), dark ice (orange triangles), weathering crust meltwater (purple diamonds) and supraglacial stream water (pink squares) samples, with ellipses representing 90% confidence intervals; and (B) loadings plot of the variables included in the PC analysis.**





**Figure 4: van Krevelen diagrams showing molecular formulae assigned (A) in all samples in the dataset; (B) in all laboratory leachates; (C) in all dark ice samples; (D) in all weathering crust meltwater samples; (€) in all supraglacial stream samples; and (F) in all weathering crust meltwater samples but not in any supraglacial stream sample. In panels B-E, formulae presented in panel A are excluded. The total number of molecular formulae displayed in each diagram is denoted in the top right of each panel, and data points are coloured according to their assigned heteroatom class with CHO compounds in black open circles, CHON in light blue squares, CHOS in orange triangles and CHONS in green diamonds. Note that only two CHONS formulae are shown in this figure, both in panel B.**

Data were plotted in van Krevelen space to visualize the molecular composition and diversity of the core supraglacial DOM pool (Fig. 4A) and the different sample types (Fig. 4B-E). The core supraglacial DOM signature, consisting of 2,885 molecular formulae that were assigned in every sample in the dataset, was comprised mainly of HUP and aliphatic compounds with the heteroatomic formula CHO or CHON (Fig. 4A). To visualize the difference between laboratory leachates, dark ice, weathering crust meltwater and stream water, molecular formulae that were assigned in all samples, but were not present in the core supraglacial DOM signature, were plotted in Fig. 4B-E, respectively. The laboratory leachate signature contained 1,509 molecular formulae, that predominantly occupied the aliphatic and HUP space in the van Krevelen diagram (Fig. 4B). The dark ice signature contained 2,577 formulae, with most formulae falling in the HUP region of the van Krevelen diagram (Fig.





4C) and a clear contribution of polyphenolic and condensed aromatic formulae. The weathering crust meltwater signature was the most molecularly diverse with an additional 3,995 formulae present in all samples, falling mostly in the aliphatic and HUP space, with ~52% of formulae containing nitrogen. The supraglacial stream signature was relatively similar to the weathering crust meltwater signature, but with fewer formulae (3,099) in all regions of the van Krevelen diagram. To further examine these differences, molecular formulae present in all weathering crust meltwater samples but not in any of the supraglacial stream samples are highlighted in Fig. 4F, where approximately 60% of the formulae fall in the aliphatic region and contain sulphur or nitrogen. Of the 282 formulae in Fig. 4F, a total of 182 formulae had also been assigned in at least one dark ice or laboratory leachate sample. Across the entire dataset, only 24 formulae were present in all supraglacial stream samples but not in any weathering crust meltwater samples.

## 4 Discussion

### 4.1 Dark ice, surface ice debris, and laboratory generated leachates

Dark ice samples analysed in this study were collected by scraping off the top 2 cm of surface ice in patches with a visible debris loading. The DOC concentrations in these dark ice filtrates (0.90 – 1.01 mg L$^{-1}$) fell within the range of values reported by Lutz et al. (2017) for 12 glaciers in Svalbard and Arctic Sweden (0.27 – 2.33 mg C L$^{-1}$), but was higher than those reported for dark ice surfaces (0.17 – 0.32 mg C L$^{-1}$) on Leverett glacier in southwest Greenland (Musilova et al., 2017). Surface ice DOC concentrations depend on local atmospheric deposition (Stubbins et al., 2012), microbial abundance and productivity (Musilova et al., 2017), mineral dust or particulate loadings (McCutcheon et al., 2021), and DOC contained within the ablating glacier surface. In addition, lysis of cells during sampling, thawing, or filtering might contribute to dark ice DOC concentrations or DOM composition. However, as lysis of microbial cells also takes place in undisturbed ice surface communities, for example as a result of fungal infections of glacier ice algal cells (Fiołka et al., 2021), the potential effect of cell lysis during sampling cannot be quantified or mitigated, but is unlikely to significantly alter the dark ice DOM composition. If cell lysis during sampling was the main source of surface ice DOM, we would expect to see similarities in dark ice and laboratory water leachate DOM composition, but these two sample groups present distinct DOM signatures (Table 1, Fig. 3A, Fig 4B-C) and we therefore assume that the dark ice DOM composition reported here is not a sampling artifact.

Surface ice debris removed during filtration of dark ice samples was used to determine TC (3.00 – 3.35 %) and TN (0.25 – 0.28 %) content, and to extract a laboratory-generated cold-water leachate for DOM characterisation. For all debris samples, TC and TN values were in the lower end of the range of values reported for ice with a high biomass loading (TC: 2.59 – 8.45 %, TN: 0.20 – 0.87 %; McCutcheon et al., 2021), likely because our sample collection took place before the peak of the typical ablation season microbial bloom and because the ice surface in the sampling region had a relatively high particulate loading (microscope observations in the field). Interestingly, laboratory leachate and dark ice DOM compositions were significantly different (Table 1, Fig. 3, Fig. 4B-C) despite originating from the same sample. Laboratory leachate DOM had a lower mean





RA weighted mass and aromaticity and a higher %RA of aliphatic and peptide-like compounds than dark ice DOM (Table 1).
The freeze-drying and cryo-milling of surface debris prior to preparation of the leachate likely lysed cells within the debris
samples, yielding DOM with a higher prevalence of aliphatic and peptide-like compounds, which are typically of microbial
origin (Kellerman et al., 2018; Spencer et al., 2015). When comparing the molecular signature of laboratory leachate and dark
ice DOM in Fig. 4B and Fig. 4C, the more aromatic signature of dark ice compared to leachate DOM is also apparent.

The lack of aromatic compounds in laboratory leachate DOM either suggests that: i) surface debris is not a source of aromatics;
ii) oxidation needs to occur to solubilize aromatics from the particulate OM; or iii) aromatics where already preferentially
leached from the surface debris before it was collected. Over 80% of the condensed aromatic formulae found in dark ice have
an O/C > 0.4, corresponding to 6.4 ± 0.8 %RA of the total dark ice DOM pool, indicating that microbial oxidation may be
taking place on the surface (Antony et al., 2017). Furthermore, Antony et al. (2017) showed that microbial reworking of DOM
in the snowpack of a coastal Antarctic site resulted in DOM with a higher aromaticity and a higher magnitude and number of
nitrogen-, sulphur- and phosphorus-containing formulae. Similar microbial reworking in surface ice may partially explain the
higher aromaticity of dark ice DOM compared to laboratory leachate, which was generated to approximate dark ice DOM
prior to microbial or photochemical alteration. A likely pathway towards higher aromaticity of dark ice DOM due to microbial
degradation is the deglycosylation of the algal pigment purpurogallin carboxylic acid-6-O-β-D-glucopyranoside, which is
classified as a HUP compound. It has previously been shown that fungal infections of glacier ice algae can cause algae to lose
their pigmentation (Fiołka et al., 2021), and that the Greenland Ice Sheet surface fungal species *P. anthracinoglaciei* is able to
convert this pigment into its aglycone derivative (Perini et al., 2023), likely utilising the sugar moiety as an energy source. The
resulting compound has the polyphenolic molecular formula $C_{12}H_8O_7$, which accounts for 5.7 – 8.8 %RA of dark ice DOM.
Atmospheric deposition resulting from biomass burning or anthropogenic activity may also deliver aromatic OM to
supraglacial surfaces (Stubbins et al., 2012). Aromatic DOM is susceptible to photochemical degradation (Spencer et al., 2009;
Stubbins et al., 2010), and recent work has shown that a 28-day irradiance experiment using burnt biomass and anthropogenic
OM leachates selectively removed aromatic compounds and produced aliphatic compounds (Holt et al., 2021). The elevated
aromaticity of dark ice suggests that photodegradation of aromatic DOM is not taking place on the ice surface, and that aromatic
compounds are both degraded and produced at comparable rates or that surface ice DOM residence times are not long enough
to achieve complete photochemical degradation of aromatics. Absence of photodegradation is unlikely given the high levels
of irradiance the surface ice receives during the ablation season. Instead, we suggest that the elevated aromaticity of dark ice
DOM is likely due to microbial reworking of surface ice debris or DOM.

**4.2 Near-surface hydrology and the DOM composition of meltwater in the supraglacial drainage system**

In the studied micro-catchment, modelled lateral interstitial flow velocity through the weathering crust was in the order of
decimetres per day, corresponding with estimates from Stevens et al. (2018), Irvine-Fynn et al. (2021) and Yang et al. (2018).
The particle track from Hole D (Fig. 2) confirmed the assumed hydrological connections between sampling locations. We



assume that water sampled from the weathering crust auger holes was comprised of a mixture of recent melt, originating near the hole, and meltwater already transiting within the weathering crust. We also assume the addition of further meltwater, percolating from the unsaturated zone of the weathering crust, along the transit pathway from Hole D to the supraglacial stream (Fig. 1E). The presence of a core supraglacial DOM signature, with 2,885 molecular formulae shared across all samples in the dataset (Fig. 4A), supports the assumption that the dark ice, weathering crust meltwater and supraglacial stream environments,

and hence DOM pools, are hydrologically connected. The indicative value of nine days for meltwater flow through the weathering crust to the supraglacial stream allows the possibility of modification of the DOM pool via microbial reworking and/or photochemical degradation (Riedel et al., 2016; Antony et al., 2017, 2018; Holt et al., 2021).

Given the hydrological connectivity of surface ice with weathering crust meltwaters, a higher degree of convergence between dark ice and weathering crust DOM may be expected. The higher concentration of DOC in dark ice relative to weathering

crust meltwater may be due to dilution of dark ice DOC in the weathering crust by meltwater produced in the subsurface and percolating from the unsaturated zone of the weathering crust, or from surface ice with a low debris loading, which likely contains less aromatic DOM. Yet, given the differences in the number and elemental composition of aromatic formulae assigned in dark ice and weathering crust meltwater, retention of DOM on the ice surface, potentially via association with extracellular polymeric substances (EPS) as suggested by Holland et al. (2019), is a more likely explanation. EPS has been

shown to play a role in the formation of granules in supraglacial cyanobacteria communities (Yallop et al., 2012; Stibal et al., 2012; Langford et al., 2010) and Perini et al. (2023) observed glacier ice algae embedded in EPS substances during co-cultivation with the surface ice fungi *Articulospora* sp. To date, the chemical composition and role of EPS in microbial communities on the ice surface (c.f. cryoconite granules) remains unknown. We recommend further characterisation of organic matter delivered by atmospheric deposition and of EPS associated with surface ice communities, combined with controlled

incubation experiments, to assess supraglacial DOM sources, retention, photodegradation, and microbial reworking on the ice surface.

In addition to the differences between dark ice and weathering crust DOM composition, the differences in heteroatomic composition, mean RA weighted mass, and the %RA of low O/C aliphatic and condensed aromatic compounds between weathering crust meltwater and supraglacial stream DOM (Table 1, Fig. 3) also point to potential microbial and/or

photochemical alteration of DOM in the near-surface. These compositional differences are unlikely to be the result of upstream contributions to supraglacial stream DOM, as only 12 formulae were uniquely assigned to all supraglacial stream samples (accounting for 0.02 – 0.05 %RA).. The high prevalence of aliphatic and peptide-like compounds, and the increase in N- and S- containing compounds relative to dark ice DOM, suggest that microbial reworking of DOM takes place within the weathering crust. The small but statistically significant decrease in %RA of aromatic DOM in supraglacial stream samples

compared to weathering crust meltwater (Table 1, Fig. 4D-F) indicates that further photodegradation of aromatic DOM might also take place during transport through the supraglacial drainage system.



## 5 Conclusions

The ice surface and the weathering crust photic zone present important sites for transformations of supraglacial DOM, altering the composition of DOM in supraglacial streams that drain to the glacier bed and deliver nutrients to subglacial and downstream

ecosystems. To our knowledge, this study presents the first characterization of DOM associated with microbial communities and weathering crust meltwater in the Greenland Ice Sheet bare ice ablation zone. The distinct composition of dark ice DOM relative to weathering crust and supraglacial stream DOM highlights the importance of future research into the role of atmospheric deposition, surface ice DOM retention by EPS, and supraglacial water residence times with regards to their effects on the resulting DOM composition in supraglacial runoff. This is particularly important as rainfall over the Greenland Ice

Sheet is becoming more prevalent (Niwano et al., 2021), consequently increasing the frequency of weathering crust degradation events (Müller and Keeler, 1969). Temporary removal of the weathering crust, which presents a site for microbial and/or photochemical alteration of DOM, could potentially result in the rapid export of more aromatic, and likely less biolabile, DOM from ice surfaces during or following rain events. Our findings have implications for the understanding of supraglacial biogeochemical cycling, emphasizing the importance of including the weathering crust photic zone when assessing

supraglacial inputs to subglacial and downstream ecosystems.

### Data availability

All FT-ICR MS data used in this study can be found in the Open Science Framework Repository via DOI 10.17605/OSF.IO/JRBTH.

### Author contributions

Study design, conceptualisation, and sample collection was done by ELD and ITS. Field processing was done by PER. DOC, TC and TN analysis was done by RA. Lab set-up for DOM extractions was done by ELD and PER. Sample preparation was done by ELD. FT-ICR MS data acquisition was done by AMM. Molecular formula assignment and classification was done by AMK. Hydrological data processing was done by ITS. Data analysis and manuscript preparation was done by ELD. All authors contributed to the final manuscript with discussion and revisions.

**Competing interests:** The authors declare that they have no conflict of interest.

### Acknowledgements

This study was financially supported by the European Research Council (ERC) Synergy Grant DEEP PURPLE under the European Union's Horizon 2020 research and innovation programme (grant agreement No 856416), the Aarhus University Research Foundation through a Starting Grant for AMA (AUFF-2018), the Aarhus University Interdisciplinary Centre for



Climate Change (iClimate), and the network programme of the Danish Agency for Science and Higher Education (9096-00101B) and the Helmholtz Recruiting Initiative grant (award # I-044-16-01 to LGB). RA acknowledges funding from the Alexander von Humboldt Foundation, and RGMS would like to acknowledge NSF DEB 1145932. A portion of this work was performed in the Ion Cyclotron Resonance User Facility at the National High Magnetic Field Laboratory, which is supported by the National Science Foundation Division of Chemistry and Division of Materials Research through DMR-2128556 and

the State of Florida. All 21 T FT-ICR MS files are publicly available via the Open Science Framework through DOI 10.17605/OSF.IO/JRBTH. Predator analysis and PetroOrg© software is publicly available for ICR facility users at https://nationalmaglab.org/user-facilities/icr/icr-software. Finally, the authors would like to thank the entire DEEP PURPLE team, especially those involved in the 2021 field campaign.

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
