# Peer review of "Molecular level characterization of supraglacial dissolved and water extractable organic matter along a hydrological flow path in a Greenland Ice Sheet micro-catchment"

_EGUsphere, 2024_

## Author Comment (AC1)

**RC1: Anonymous Referee #1, 05 Apr 2024**

Dear editor and reviewer,

We would like to thank reviewer #1 for their constructive comments and suggestions. Please find our response to each comment (in blue) below. We appreciate the time and effort from the editor and all three reviewers, and hope that our responses will be sufficient to be allowed to submit a revised manuscript for publication in Biogeosciences.

Overall General:
- Please introduce reserach questions and hypothesis that guide throught the text
  We agree with the reviewer that the research questions and hypotheses should be made clearer throughout the text and will focus on this when revising the manuscript.

Abstract
- Title is missing a preposition in the second line connecting the southern Greenland Ice Sheet
  It seems that the word "on" disappeared from the preprint .pdf file, while it does show up in the online version of the preprint. In the revised version, the title will be updated to "Molecular level characterization of supraglacial dissolved organic matter in a hydrologically connected Greenland Ice Sheet micro-catchment" or similar - in line with changes during manuscript revision and following a comment from reviewer #3 regarding the use of the term 'exported pools'.

- L 18: Make sure readers not acquainted with glacial dynamics can follow your text, you could specify that the ablation season is connected to the boreal summer months
  This will be made clearer in the revised manuscript.

- L 18: Are bacteria only more active during that time or are the growing and colonising, the statement you give is ambiguous
  It has been shown that they are metabolically active and are growing (increasing in number and spatial coverage) during the ablation season - but data from other seasons are currently lacking. This sentence will be rephrased in the revised manuscript.

- L19: "the DOM may be exported", was there any evidence that DOM is actively exported from the supraglacial ice, if so then you should use a stronger verb
  Yes, there is evidence in the literature that supraglacial DOM is exported, and this sentence will be revised accordingly.

- L24: What is meant here by supraglacial dark ice: biologically darkened ice? You should help readers not acquainted with all processes and particularities by taking them by the hand, especially for the abstract.

We indeed intended this to refer to biologically darkened ice but agree that this should be specified upon first mention. We will ensure that jargon is explained more thoroughly throughout the revised manuscript.

Introduction

- General point: I am missing a general research question and hypothesis deducted here. Please try to make a clearer statement on your scientific goal. It would be nice to also have this question(s) mentioned in the abstract
  We agree that the guiding research question and hypothesis got somewhat diluted during the internal revision stages. In the revised manuscript, we will make the general research question and hypotheses more explicit in the abstract and introduction and will refer to them directly in the conclusion.

- L59: Since aromatic composition DOM was found in glacial run-off, not all of the degradation of allochthonous aromatic DOM can happen during the transport to glacier surfaces.
  Agreed - this will be rephrased in the revised manuscript.

- General Comment Introduction: Indeed, glacial runoffs are known to show a "dual" source with both aromatic allochthonous and biolabile autochthonous DOM sources. Potentially the aromatic allochthonous fraction would be expected to be more susceptible to photodegradation, while the autochthonous fraction more susceptible to biodegradation. Did you find any evidence of production of Carboxyl-Rich Alicyclic Molecules (CRAM) from algal DOM in any other studies of glacial ecosystems (14-C young aromatics then), this could help to decipher that parts of the allochthonous aromatic DOM could in fact be autochthonous as well. Did you investigate 14-C ages in your samples? If not you can have a specific look on the polycondensed aromatic fraction in your FT samples to gain insight if a high number of black carbon-like, likely allochthonous molecular formulae exist in your samples, if so they could likely be highly susceptible to photodegradation.
  We have not assessed CRAM in our data - and did not specifically evaluate black-carbon-like formulae, but report condensed aromatic %RA. In depth evaluation of supraglacial black-carbon material is currently lacking and is the topic of ongoing and future research. Supraglacial radiocarbon data are limited to supraglacial streams and range from -350 to 23 ‰, and to our knowledge radiocarbon data associated with supraglacial algae or algal DOM do not yet exist. We will include a sentence on the susceptibility of autochthonous OM to photodegradation in the revised manuscript to address this.

- L70: Upon draining of DOM to the glacial bed where any studies performed that investigated how supraglacial DOM was transported in the glacial bed, were there any interactions and exchanges? I guess it is unlikely that supraglacial DOM behaves inert when passing englacial and subglacial systems. You could also review these sources.

To our knowledge there are no studies investigating the transformations of DOM en route from the glacier surface to the glacier bed or to the glacier terminus, likely due to the difficulties in accessing the englacial and subglacial environment. Yet, given the previously reported differences between supraglacial stream DOM and glacial runoff DOM, it is reasonable to assume that subglacial processes change the DOM composition through additional inputs and degradation. This will be made clearer in the revised manuscript.

Materials and Methods
- L83 please define "small" by estimate or accurate metrics
  Dimensions will be added in the revised manuscript.

- L85 was the auger hole freshly produced? How long did it take for auger holes to refill?
  The auger hole was drilled in at 07:00 local time on July 28, 2021. Refill rates varied throughout the day, ranging from 24 seconds to 117 minutes.

- L90 times given are local time? Why was not timestamp given for the sampling of Q
  Correct, these were in local time, which will be made clearer in the revised manuscript. Sampling of Q also took place at 14:00 - we understand that phrasing was unclear, and this will be updated in the revised manuscript.

- L90 it should already be clearer what this Q is without having to inspect Figure 1
  Agreed - this will be made clearer in the revised manuscript.

- L93 last comma replaced by "and"
  This sentence will be edited for clarity in the revised manuscript.

- L94 I wonder that SPE samples were stored in simple PC bottles, while DOC aliquots were stored in PTFE / glass bottles. The SPE samples will be considered more delicate than the DOC aliquot, could you elaborate on this. Did you conduct process blanks for SPE samples with FT analysis (also including the GF filtering procedure)?
  We stored the samples for DOM extraction in acid-cleaned polycarbonate bottles as these have been shown to produce minimal DOC contamination (S0043135498004072) and are lighter and less fragile than glass bottles. Given the limited sample volume available and the amount of sample required for DOM extractions, we could not collect the DOM samples in duplicates and therefore it was essential to prevent breakage of sample containers. As DOC analysis requires much less volume than DOM extraction (in the case of glacial samples), we were able to collect duplicate samples to minimize potential sample loss as a result of glass breaking in transit from the ice camp to the laboratory in Germany. We used MilliQ water to conduct a process blank for DOM, including filtering and extraction, and formulae with a relative abundance >0.1% in the procedural field blank were removed from the dataset (see L182 in section 2.8).

- L96 replace "home" by something more clear (where)

This will be updated to the specific laboratory (GFZ Potsdam).

- L96 Which analysis did filter retentates (surface debris) undergo, it came out of the blue in the M&Ms and results should be mentioned in the abstract and POM should be discussed in the introduction
POM is not included in the introduction as we did not perform analysis on the POM. We only used the surface debris to generate a cold-water extract to represent the portion of DOM that might feasibly be released from the surface debris on the ice sheet surface, where it is in contact with cold water. We will make this clearer in the abstract and introduction of the revised manuscript.

- Figure 1: Minimap (A) has weak contrast, the box study area is not well readable, northing information is missing. Statement on the used basemap is missing; Map B is fairly too zoomed in, it doesn't become clear what is shown (valley, slope, hill crest) maybe use the drone image underlain by additional geomorphologic information that make the sampling information clearer; The "field site" was not found by me on any of the maps; The categorisation of ice types shown in the legend does refer to illustration F only (?), how can you suspect this illustration to be true, is it clear that the depth of layers has roughly these dimensions- was ground trothing carried out by digging a snow / ice profile? Dimensions in panes C,D,E and F are missing.
The color of the map will be changed in the revised manuscript. The field site is marked in panel A by a black square and coordinates are provided in section 2.1 as the use of additional information in Figure 1 would limit readability, but we will add additional information to the figure caption for clarity. A digital surface model to show 'valley, slope, hill crest' was not included for clarity and conciseness of the figure, but it may be added in the revised manuscript or revised supplementary information. Scale bars will be added to panel C-E. Panel F is an exemplar schematic which intentionally does not include scales – the depths of the layers are variable and were not confirmed (although water table height is included in the results and hydrological modeling). No ice or snow pits, note that this is the snow free ablation zone, were dug. Panel F simply presents a conceptual model to help the reader visualize the supraglacial hydrology discussed in the manuscript.

- L 108: Calculations based on Stevens 2018: please elucidate more on this
These calculations are described only briefly in this manuscript as they are a) already published, b) described in the supplementary information and c) would add considerable length to the manuscript detracting from its main research story. We believe that suitable reference is provided for interested readers to further explore the detail, nuance, and evaluation of the point hydrological methods used - an approach which was also applied by Stevens in the following papers, where it was deemed acceptable: s43247-022-00609-0 and s41467-021-24040-9. We will add a clearer reference to the supplementary information, which is currently located at the end of this paragraph, in the revised manuscript.

- L118: Can you elucidate from the literature how Milli-Q extracts might influence the DOM yield compared to other extraction techniques
  Some of the co-authors are currently working on POM extractions that include other extractants. However, we did not include a discussion on this in the manuscript as we, for this dataset, are only interested in the DOM that might be 'extracted' from surface debris by supraglacial meltwater (cold water) alone, rather than in a full characterization of all DOM that might be extracted from surface debris using more elaborate chemical methods that do not represent environmental conditions.

- L 120: "wwPTFE" = PTFE |Typo?
  This refers to water-wettable PTFE filters as stated on the filter packaging. We will change it to hydrophilic-PTFE to in the revised manuscript to avoid confusion (most PTFE is hydrophobic).

- L123: Usually such products are called water extracted organic matter (WEOM) "laboratory leachates" could also be column leachates asf.
  We will change this to water soluble organic matter (WSOM) in the revised manuscript.

- L128: The high temperature combustion technique doesn't need to be described with this detail, it appears to be a standard
  This section will be condensed in the revised manuscript.

- L 136 – 137 see comment on L128
  This section will be condensed in the revised manuscript.

- L 141: see comment L 96
  This will be updated to the specific laboratory (GFZ Potsdam).

- L 149: please indicate what these supplementary methods are roughly about
  This will be elaborated on in the revised manuscript.

- L160 and L163: Since Equation 1 and 2 are standard and published, citations will suffice instead of formulae
  This section will be condensed in the revised manuscript.

- L 165 f.: I highly recommend changing the naming of your "composition groups" to a less ambiguous naming. The current naming implies structure which cannot be determined by mass spectrometry. A less ambiguous nomenclature is presented by Merder et al. 2020 (https://dx.doi.org/10.1021/acs.analchem.9b05659) Table 2 in the Supplements, which is also co-authored by some of the Co-Authors here.
  Most studies presenting glacial DOM data relevant to our manuscript use these classifications, enabling some degree of comparison between our data. The current naming is not intended to imply structure (e.g. peptide-like) and is broadly considered

conservative. We will make it clearer that we refer to '-like' formulae in the revised manuscript as to avoid ambiguity regarding structural information.

- L 171: This sentence can be deleted; I expect you wouldn't act in the opposite way. If this is true on the other hand, is not proven by your sentence, you would need to argue this in the introduction justifying your selection of metrics by citations. Further metrics could be tested such as IDEG (Flerus et al., 2012) or ITERR (Medeiros et al., 2016) and the investigation of pcARO could also function as metric/
This section will be updated to reduce ambiguity and include the relevant citations.

- L 176/177: Can you present any insight into the actual biolability of purpurogallin as an empiric measure for your discussion. It doesn't become clear to me what you imply here Example: An essential amino acid like tyrosine contains an aromatic ring structure and is widely considered biolabile: If the classification of biolability is closely connected to refractory characteristics of structures, this discussion capsizes when conditions of biological decomposition are not made clear or no empirical landmark on actual degradability is given.
This section will be updated to reduce ambiguity and include relevant citations, including to Perini et al 2023 (s00248-022-02033-5 ) who report the biodegradation (removal of the sugar moiety) of the purpurogallin pigment in fungal incubations.

- L 180 2.8 Statistics: Besides homoscedasticity did you test for normal distribution and were the sample sizes evenly distributed. With heteroscedasticity, non-normality and uneven samples sizes (which is the common case in geosciences) metric testing becomes less and less trustworthy. Please add normality testing, histograms and sample sizes to your supplements. In case of multiple violations of prerequisites for metric testing consider either multiple non-parametric testing e.g. with package "multcomp" (Bretz et al. 2011 ISBN 9781584885740) or Box-Cox transformation of data prior to ANOVA
Normality testing, histograms and sample sizes will be added to the supplements of the revised manuscript as suggested by the reviewer.

- Figure 2 and section 3.1: Figure please add DEM data to the figure, here the viewer can just see blurry white with a large pixel size. I also wonder, what is the margin of error for your 9 days of travel time. The resolution of the orthophoto indicates that there are several potential travel passes that might occur for a single particle with certain likeliness
This figure demonstrates velocity of water flow in the weathering crust. A DEM represents the bare earth surface only, not natural features (such as an ice sheet) and would not add relevant information to Figure 2. Adding a DSM would also be redundant here as a combination of the water table and hydraulic conductivity of the weathering crust determines water flow paths along the hydraulic gradient, and this is is not necessarily the same as surface slope which would be shown by a DSM. We can include a DSM and water table map in the supplementary information if needed, but in our opinion these should not be in the main text as they are data that contribute to the

final processed product presented in Figure 2. Regarding uncertainty, the hydrological modeling approach used combined with unrepeatable point measurements means that we are not able to provide a meaningful uncertainty estimate, and these data should be viewed in that context.

- L 230: I like how you follow these single formulae through your dataset, I imagine it might be interesting to produce a figure from this finding and also underlie it with some of your matching metrics. Since purpurogallin should absorb light, maybe you could also add some UV absorbance values if you still have some back-up sample to analyse- these might match. The current way of pure text and numbers presentation is making it hard to follow these exciting insights
  This is a great suggestion, and we will aim to include a figure to summarize these findings in the revised manuscript. Unfortunately, we do not have any sample available for UV absorbance analysis, but we agree that this would be a great addition to future work.

- L 239: There are dozens of definitions on diversity, please specify which diversity you refer to
  In the revised manuscript we will avoid the use of 'diversity' and will stick to 'number of formulae' to avoid confusion.

- L 249: Lettering indication should be self-explanatory please delete starting from "where values that have…."
  This will be updated in the revised manuscript.

- Figure 3: There are n= 17 samples and n= explanatory variables. The low ratio of sample to variable (1.21) suggests that the PCA model is not as selective as it could be. Please check your variables for collinearity and make sure to remove collinear variables. I am missing the Eigenvalue {Variance} / Component documentation, please add this to your supplementary data.
  We will double-check this and be sure to include details, including Eigenvalue {Variance} / Component documentation in the supplementary information of the revised manuscript.

- Figure 4: The pane lettering in A-F is necessary but it would help the readership to also name the sample type above the pane. Since van Krev. plots are always prone to overplotting please consider scaling the point size to %RA. You could also specify in the plot what is % RA of the shown formulae versus the excluded formulae to specify not only diversity but also intensity
  We will update the pane titles to be clearer. However, in our experience the scaling of point size to %RA causes overplotting to be worse instead of better and was therefore not used in presenting this dataset.

- Figure 4 vs Table 1: how do values in Table 1 correspond to Figure 4: Since you conducted a perfectly interesting subtraction technique for Figure 4 it would be nice to

append a table 2 with the respective metrics (as indicated van Krev. are overplotted and usually not as insightful)
We will update Table 1 accordingly in the revised manuscript.

- L 276 there is a € instead of (E)
  Thank you for spotting this - this will be corrected in the revised manuscript.

Discussion
- General: In the discussion you present a large number of metric data (x +- y % ) asf. Please try to limit these numbers to an absolute minimum and rather state the trends and significant differences from your results by rephrasing them in words. This will make it easier to follow the arguments. Also please try to discuss one thought in one paragraph only. It might be a good idea to enter subheadings above paragraphs to make clear which idea is discussed at the moment
  We agree that the number of metric data in the text should be reduced, and will streamline the general structure of the discussion in the revised manuscript.

- 4 Discussion: Instead of descriptive heading for 4.1 and 4.2 I would be very happy if you could include your research questions into the headings
  This will be incorporated in the revised manuscript.

- L 309: The whole section about the problem of lysis is too prominent in my eyes. You use very much space to discuss a potential artefact that you then rule out in the end did probably not happen at all or have no effect. Please shorten here. The readership should learn more about what the data tell you about potential processes here
  This section will be condensed in the revised manuscript.

- L338 how impactful can the two described degradation pathways of viral infection and fungal attack be. Undoubtly, they will have effect on the composition, but I would expect the effect to be much smaller. Also if I understand correctly, you identified the sample by the algal pigmentation visible as dark ice? So there was no tremendous viral induced loss of pigmentation
  We suggest that fungal infection can result in the loss of pigment from algal cells (note that there is no current evidence of viruses attacking glacier ice algal cells and we do not suggest this as a mechanism). Work by Fiołka et al (2021, s41598-021-01211-8#Sec22) investigated a biologically darkened ice surface and found that approximately 25% of all cells in the sample collected from this biologically darkened site were infected with parasitic fungus. Hence, we argue that it is reasonable to expect a contribution of pigment leaching to the surface ice DOM pool. We will rephrase this argument for clarity in the revised manuscript.

- L340: The NOSC metric could be used to maybe hint into the same direction.
  This will be considered in the revised manuscript.

- L344: A similar study was following DOM from source to sink in a closed alpine system. there is clear indication of photodeg Part 4.3 https://doi.org/10.5194/bg-20-3011-2023
  Interesting! We will include this reference in the revised manuscript.

- L345: Especially dark ice could also shield lower lying aromatics from sunlight and subsequent photodegradation by the low albedo of overlying aromatics. This is also shown for ocean darkening by various indicators
  https://doi.org/10.3389/fmars.2020.547829
  Yes, shielding is likely to also play a role in the supraglacial ecosystem.

- L 350 I would also suggest shielding of underlying aromatics
  This will be included in the revised manuscript.

- L362 here it would of course have been nice if you had carried out some photodegradation experiments with your samples to track this pathway
  Agreed! We hope to include photodegradation experiments in future studies to improve our process understanding of biogeochemical cycling during supraglacial transit.

- L365 here also lysis products might accumulate
  Correct - this will be included in the revised manuscript.

- L369 treat EPS as plural pls.
  Thank you for spotting this - this will be updated in the revised manuscript.

- L 373, that is an important recommendation, but it is very big in the light what a single paper can achieve, that why it would be better to ask this in the form of questions and to mark knowledge gaps more precisely
  Great point - we will make the specific questions to be addressed more explicit in the revised manuscript.

- L382 but the, if the stream is not sharing a large amount of DOM composition and not showing mixing, how can you attribute it to be a connector of pools as happened in 3.3, then the outcome must be that pools are distinct and not connected by continuous flow, which can make sense with low flow velocities
  This is incorrect. The stream does share a large number of formulae (see Figure 4). The 12 formulae mentioned in L382 are the only formulae that were assigned in all stream samples, but not in any other samples (i.e. uniquely assigned).

- L382 double ".”
  Thank you for spotting this - we will fix this in the revised version.

- L 377 f., the last paragraph comes without any reference to other scientific works, please try and also discuss these findings in the light of existing literature

The discussion needs some restructuring to focus on one argument per paragraph as mentioned in a RC above. We will align the discussion with the research question and hypothesis more explicitly in the revised manuscript.

- General question: how do you assess the different contributions of bio- and photodegradation in your sample set. I would like to see a clearer statement on which pools are to what extent affected by what
  We will make this distinction clearer and more explicit in the revised manuscript.

Conclusion
- General: I would prefer to also see a connection to research questions in this chapter
  We aim to make the research question and hypotheses more explicit in the abstract and introduction and refer back to them directly in the conclusion.

- L 390 what exactly do you mean by "microbial communities" this implies that micros where somewhat investigated more closely then presented here
  We will rephrase this to be in line with what is presented in this dataset for the revised manuscript.

- L 392: You state distinct composition differences; this is where you should say what exactly you found instead of suggesting more research
  Agreed - we will update this in the revised manuscript.

- L 394 and 395 The citations of Niwano and Müller,Keeler could also be moved to the introduction, this rather seems to be a motivation for your study than something relevant to the conclusions.
  Agreed - we will move these citations to the introduction in the revised manuscript.

---

## Author Comment (AC2)

**Responses to RC2: Muhammed Fatih Sert, 05 Apr 2024**

The paper by Doting et al. reveals the DOM molecular composition of a supraglacial micro catchment in the Greenland ice sheet surface with FT-ICR MS analysis. The manuscript is well-written and presents a unique dataset that holds significant value for the scientific community. However, the dataset utilized in the manuscript is solely limited to DOC concentrations and MS analysis and does not provide a comprehensive overview of environmental biogeochemistry in the studied site. Therefore, I would highly recommend to the authors that they may seek additional measurements to document environmental variabilities such as nutrients, isotopic compositions, or microbial diversity. If this is not possible for this manuscript, you should acknowledge this limitation by discussing the potential implications of the dataset's scope and suggesting areas for future research. Additionally, I noticed that the manuscript suffers from complicated nomenclature, which implies different meanings for researchers from other disciplines. Therefore, I would suggest that the terms used in the manuscript be carefully reconsidered for alternatives (see below).

Dear editor and reviewer,

We would like to thank the reviewer for their constructive comments. Please find our responses below (in blue). We agree that additional biogeochemical data on the sample would have been of added value to this study. However, unfortunately this was not within the scope of the study and we do not have samples that can be analyzed for the proposed parameters. We agree that this would be an important avenue for future research, and will acknowledge this limitation in the revised manuscript. In addition, we will revise nomenclate where necessary, and will ensure that any field-specific terminology is defined at first mention throughout the manuscript. We appreciate the time and effort from the editor and all three reviewers, and hope that our responses will be sufficient to be allowed to submit a revised manuscript for publication in Biogeosciences

Please see below for additional comments:
Line 36: if it provides protection from the UV, then it should reflect more and elevate the albedo rather than lowering it.
This is incorrect. The main pigment in Ancylonema nordenskiöldii and Ancyloname alaskanum is purpurogallin carboxylic acid-6-O-β-D-glucopyranoside, as identified by Remias et al., 2012. That same study reports the extensive absorption capacity of this pigment over a wide range from UV B to VIS radiation, suggesting that this phenolic compound shields chloroplasts against a surplus radiation, preventing photoinhibition during high irradiance. As protection is provided via absorption, and the pigment is in part responsible for the biological darkening cause by algal growth on the ice sheet surface, its presence lowers the albedo (the fraction of light that is reflection by a body or surface) as demonstrated by e.g. Cook et al 2017, Williamson et al 2019 and Cook et al 2020.

Line 91: Dark ice resembles ice that is not in contact with the sunlight, which is not the case. You may consider using a different term. Maybe brown ice or just surface ice.
In the revised manuscript, we will make sure to introduce this term clearly or change to biological-darkened ice. Within the cryosphere community, dark ice is commonly used to describe ice surfaces with dust, soot or microbial loading (for example, when searching "dark ice" on Google.com, the top science result describes dark ice in the context of ice sheets and glaciers) - hence, this is the appropriate term to use in the context of this paper.

Line 118: Leachate does not really define what you obtained from your samples. Leachate usually defines liquids that are drained through solids gravitationally or maybe via osmosis. What you did is more like an extraction rather than leaching because of 500 rpm shaking and centrifuging. Therefore, I would use a different term, such as surface debris extract. On the other hand, the component you define as the weathering crust is more like leachate because you let the auger hole fill with meltwater leachate and perform the sampling afterwards.
In the revised manuscript, we will replace 'laboratory leachate' with water soluble organic matter (WSOM).

Section 3.3: PCA with dependent variables does not reveal the compositional differences between samples. It is not surprising that the CHON, CHO and CHOS point in different directions on the plot because they are basically the opposite representation of the same variable. The same applies to formula percentages of van Krevelen regions. PCA is usually for independent environmental variables to indicate how environmental conditions differ for the sampling sites. You should consider applying clustering methods on relative intensities for compositional differences between samples then you can add parameters on the ordination plot (e.g. NMDS + envfit in Vegan). You would possibly get a similar separation, but then you may know which cluster of samples is more similar to the other cluster compared to the remaining ones. Then, instead of PCA, you may simply use bar plot or box plot to visualize which parameter infer the bigger variation.
Principal component analysis is commonly used to assess the DOM parameters that distinguish FT ICR MS samples, (e.g. in Riedel et al 2016, Spencer et al 2019, Kellerman et al 2021, Marshall et al 2021, Holt et al 2021) and we therefore consider it an appropriate method to examine our dataset and compare with existing literature. The data suggested for the bar plots are presented in Table 1. We selected a table over bar plots to keep the number of figures in the manuscript to an appropriate number.

Figure 4: The figure caption is unclear, and I did not understand what the individual plots show. You should extend this figure by adding van Krevelen for all the formulas obtained from each sample. For example, 8403 formulas for lab leachate or 7540 formulas for dark ice. You could also involve RAs by symbol sizes.
We will update the figure and figure caption to be clearer in the revised manuscript.

Line 320: What do you mean by high particulate loading? What kind of particulate matter? You may explain more about what you observed with the microscope in the field.
We will elaborate on our observations in the revised manuscript.

Line 326: why preferentially? Are those more soluble?
We assume this is meant to refer to L329. With 'preferentially' here we mean DOM that has already leached from the surface debris, and will therefore not leach from the material again when preparing the water extract. We will rephrase this to make it clearer in the revised manuscript.

Line 360: You should have check how molecular intensity of common molecular formulas change between samples. You have done this for several selected formulas, but you could extend this to all common formulas to see if there is indeed hydrological connectivity. Otherwise, number of common formulas do not necessarily indicate hydrological connection. You would have found common molecular formulas in any DOM samples.
Common formulae alone don't demonstrate hydrological connectivity, but paired with the hydrological data, which suggest hydrological connectivity, the large overlap in shared formulae strengthens the hydrological connectivity argument. Looking at changes in molecular intensity therefore seems redundant and not in line with reviewer requests to streamline the paper.

---

## Author Comment (AC3)

**RC3: Anonymous Referee #3, 18 Apr 2024**

The authors of "Molecular level characterization of supraglacial dissolved organic matter sources and exported pools the southern Greenland Ice Sheet" present a research project on DOM composition from various sources and hydrologic flow of material to the coast of southern Greenland. The goal of the work relies on molecular composition comparisons of samples collected in the field and leached in the lab. The work describes themes of DOM transformations on the Greenland ice sheet and during downstream transport but is organized in a way that is confusing. Some of those points are noted as important (for example in the introduction) and language focuses on assessing potential transformations of DOM from sample collections, but some connections between results and assessing these transformations of DOM are confusing throughout the text, and some parts may be missing (for example % biolabile values are included in the abstract, but not reported on in the main text). Maybe the introduction can be reorganized to narrow the scope/set the foundation for what's to come a bit better, creating a clearer start, and the rest of the manuscript text follows with more clarifying text so that the messages are clear and continuous throughout. The language describing the importance of microbial transformations and photochemical transformations in the introduction is not paired well with the lack of microbial and photochemical data for this project. It reads like something is missing. The first two paragraphs of the introduction create a wide scope of the work, which is not supported by the rest of the text. Connecting those ideas across the introduction and results sections will be greatly strengthened and reduce confusion. All recommendations for revision can be achieved. The following comments are divided into two groups, major and minor revisions.

Dear editor and reviewer,

We would like to thank the reviewer for their constructive comments and suggestions. Overall, we agree that the manuscript needs to be streamlined to make the research questions, hypotheses and major findings clearer and easier to follow. Please find our responses to each individual comment below (in blue). We appreciate the time and effort from the editor and all three reviewers, and hope that our responses will be sufficient to be allowed to submit a revised manuscript for publication in Biogeosciences

Major Revisions
1. The fluctuating use of the following words, compounds, molecular formulae, and composition, needs to be corrected for consistency and clarity. Parts of the introduction and results section use these terms interchangeably, yet they each have different meanings and may point to different measurements. It is not clear if the introduction is discussing composition measurements of DOM from various instruments or are they all FTICRMS? Are some of these studies measuring aliphatic compounds directly?
   We will correct terminology throughout the manuscript for consistency and clarity and will make methods by which DOM in other studies was characterized explicit in the introduction.

2. Introduction: Confusing themes, see some examples already stated in the first paragraph of this review. The first two paragraphs of the introduction seem like they should be the second and third paragraphs of the section and an opening paragraph should be added that sets the stage. The first sentence of the introduction focuses the reader on microbial blooms. Is that the most important thing to start with? Why? The same type of comment is true for the beginning of the second paragraph.
We agree with the reviewer that the scope of the introduction is too broad and should be edited to more explicitly introduce the most important themes, the main research question and the main hypotheses. We will improve this in the revised manuscript.

3. Some text in the abstract, introduction, and conclusion states more than what experiments were conducted and what was measured. Please clarify the language to be more specific and reduce confusion. State where you are speculating. Some text reads as though you monitored the transformations of DOM during downstream transport. This is a major source of confusion. Please clarify. Example in Line 74, "as it is transported" suggests that you followed a parcel of water and made collections during transport. Is that true? Were they grab samples along a gradient?
In the revised manuscript, we will rephrase statements to remove ambiguity and focus on the main findings of the dataset. We feel that the edits suggestions in point 2. will help set up the manuscript so that this will be easier to achieve.

4. FTICRMS details in the methods are missing. What ionization technique and mode were used? If one type of ionization and mode were selected, why? Were your ions singly charged? If so, how did you confirm that? This is especially important when going from m/z values to masses in Da. What was the signal/noise threshold? How were molecular formulae assigned? These are important details to include in the main text. The supplementary section doesn't provide enough details either and repeats some of what is included in the main text. This work would also benefit from reporting on their instrument performance from their DOM optimization standards, see Hawkes et al., 2020, especially since they are reporting on AImod, m/z, H/C, and O/C metrics from their samples. Are there instrument biases?
Reference: Hawkes, J.A. et al. 2020, An international laboratory comparison of dissolved organic matter composition by high resolution mass spectrometry: Are we getting the same answer? Limnology and Oceanography Methods, 2020, 18: 235-258.
We will revise and update the methods and supplementary methods section in line with reviewer suggestions. See below for a more elaborate 21T method description, which will be further edited and elaborated in the revised supplementary information.

Instrumentation: ESI Source.
Sample solution was infused via a micro electrospray source[1] (50 μm i.d. fused silica emitter) at 500 nL/min by a syringe pump.  Typical conditions for negative ion formation were: emitter voltage, -2.8-3.2 kV; S-lens RF level: 40% ; and heated metal capillary temperature, 350 ° C.

Instrumentation: 21 T FT-ICR MS.

DOM extracts were analyzed with a custom-built hybrid linear ion trap FT-ICR mass spectrometer equipped with a 21 T superconducting solenoid magnet.[2,3] Ions were initially accumulated in an external multipole ion guide (1-5 ms) and released m/z-dependently by decrease of an auxiliary radio frequency potential between the multipole rods and the end-cap electrode.[4] Ions were excited to m/z-dependent radius to maximize the dynamic range and number of observed mass spectral peaks (32-64%),[4] and excitation and detection were performed on the same pair of electrodes.[5] The dynamically harmonized ICR cell in the 21 T FT-ICR is operated with 6 V trapping potential.[4, 6] 100 individual Time-domain transients with AGC ion target of 2E6 charges per scan[AM1] of 3.1 seconds were conditionally co-added and acquired with the Predator data station that handled excitation and detection only, initiated by a TTL trigger from the commercial Thermo data station, with 100 time-domain acquisitions averaged for all experiments.[7] Mass spectra were phase-corrected [8] and internally calibrated with 10-15 highly abundant homologous series that span the entire molecular weight distribution based on the "walking" calibration method.[9] Experimentally measured masses were converted from the International Union of Pure and Applied Chemistry (IUPAC) mass scale to the Kendrick mass scale[10] for rapid identification of homologous series for each heteroatom class (i.e., species with the same $C_cH_hN_nO_oS_s$ content, differing only be degree of alkylation).[11] For each elemental composition, $C_cH_hN_nO_oS_s$, the heteroatom class, type (double bond equivalents, DBE = number of rings plus double bonds to carbon, DBE = C –h/2 + n/2 +1)[12] and carbon number, c, were tabulated for subsequent generation of heteroatom class relative abundance distributions and graphical relative-abundance weighted images and van Krevelen diagrams.[13] Peaks with signal magnitude greater than 6 times the baseline root-mean-square (rms) noise at m/z 400 were exported to peak lists, and molecular formula assignments and data visualization were performed with PetroOrg © software.[14, 15] Molecular formula assignments with an error >0.5 parts-per-million were discarded, and only chemical classes with a combined relative abundance of ≥0.15% of the total were considered.

**References**

1.      Emmett, M. R.; White, F. M.; Hendrickson, C. L.; Shi, S. D.-H.; Marshall, A. G., Application of micro-electrospray liquid chromatography techniques to FT-ICR MS to enable high-sensitivity biological analysis. J. Am. Soc. Mass Spectrom. 1998, 9, (4), 333-340.

2.      Hendrickson, C. L.; Quinn, J. P.; Kaiser, N. K.; Smith, D. F.; Blakney, G. T.; Chen, T.; Marshall, A. G.; Weisbrod, C. R.; Beu, S. C., 21 Tesla Fourier transform ion cyclotron resonance mass spectrometer: A national resource for ultrahigh resolution mass analysis. J. Am. Soc. Mass Spectrom. 2015, 26, 1626-1632.

3.      Smith, D. F.; Podgorski, D. C.; Rodgers, R. P.; Blakney, G. T.; Hendrickson, C. L., 21 Tesla FT-ICR mass spectrometer for ultrhigh resolution analysis of complex organic mixtures. Anal. Chem. 2018, 90, (3), 2041-2047.

4.      Kaiser, N. K.; McKenna, A. M.; Savory, J. J.; Hendrickson, C. L.; Marshall, A. G., Tailored ion radius distribution for increased dynamic range in FT-ICR mass analysis of complex mixtures. Anal. Chem. 2013, 85, (1), 265-272.

5.      Chen, T.; Beu, S. C.; Kaiser, N. K.; Hendrickson, C. L., Note: Optimized circuit for excitation and detection with one pair of electrodes for improved Fourier transform ion cyclotron resonance mass spectrometry. Rev. Sci. Instrum. 2014, 85, (6), 0666107/1-066107/3.

6.      Boldin, I. A.; Nikolaev, E. N., Fourier transform ion cyclotron resonance cell with dynamic harmonization of the electric field in the whole volume by shaping of the excitation and detection electrode assembly. Rapid Commun. Mass Spectrom. 2011, 25, (1), 122-126.

7. Blakney, G. T.; Hendrickson, C. L.; Marshall, A. G., Predator data station: A fast data acquisition system for advanced FT-ICR MS experiments. Int. J. Mass Spectrom. 2011, 306, (2-3), 246-252.

8. Xian, F.; Hendrickson, C. L.; Blakney, G. T.; Beu, S. C.; Marshall, A. G., Automated Broadband Phase Correction of Fourier Transform Ion Cyclotron Resonance Mass Spectra. Anal. Chem. 2010, 82, (21), 8807-8812

9. Savory, J. J.; Kaiser, N. K.; McKenna, A. M.; Xian, F.; Blakney, G. T.; Rodgers, R. P.; Hendrickson, C. L.; Marshall, A. G., Parts-Per-Billion Fourier transform ion cyclotron resonance mass measurement accuracy with a "Walking" calibration equation. Anal. Chem. 2011, 83, (5), 1732-1736.

10. Kendrick, E., A mass scale based on CH2 = 14.0000 for high resolution mass spectrometry of organic compounds. Anal. Chem. 1963, 35, (13), 2146-2154.

11. Hughey, C. A.; Hendrickson, C. L.; Rodgers, R. P.; Marshall, A. G.; Qian, K., Kendrick Mass Defect Spectroscopy: A Compact Visual Analysis for Ultrahigh-Resolution Broadband Mass Spectra. Anal. Chem. 2001, 73, 4676-4681.

12. McLafferty, F. W.; Turecek, F., Interpretation of Mass Spectra, 4th ed. University Science Books: Mill Valley, CA, 1993.

13. van Krevelen, D.W. (1950). "Graphical-statistical method for the study of structure and reaction processes of coal", Fuel, 29, 269–84.

14. Kim, S.; Kramer, R.W.; Hatcher, P.G. Graphical Method for Analysis of Ultrahigh-Resolution Broadband Mass Spectra of Natural Organic Matter, the Van Krevelen Diagram. Anal. Chem. 2003, 75, 20, 5336–5344

15. Bahureksa, W.; Borch, T.; Young, R.B.; Weisbrod, C.; Blakney, G.T.; McKenna, A.M., Improved Dynamic Range, Resolving Power, and Sensitivity Achievable with FT-ICR Mass Spectrometry at 21 T Reveals the Hidden Complexity of Natural Organic Matter, Anal. Chem., 94 (32), 11382-11389 (2022.)

16. Corilo, Y. E. PetroOrg Software, Florida State University, Omics LLC: Tallahassee, Fl, 2014.

Minor Revisions

Title: Typo. There is a word missing in between "pools" and "the". Is the word "sources" necessary? Is "exported pools" a bit farfetched for this work? The work describes a supraglacial microcatchment, so perhaps "exported pools" is too wide a phrase.

It seems that the word "on" disappeared from the preprint .pdf file, while it does show up in the online version of the preprint. In the revised version, the title will be updated to "Molecular level characterization of supraglacial dissolved organic matter in a hydrologically connected Greenland Ice Sheet micro-catchment" or similar - in line with changes during manuscript revision and following this comment.

Abstract: There is one sentence of results in this abstract and then the following sentence states how these findings have implications and importance for future work. Not enough information about the results of the work and why it is important is included in the abstract.

We will revise the abstract to include more results and will elaborate on consequences of those results.

Introduction:

Lines 43-44: Not true. This has been characterized in Antarctica. Please be more specific.

To our knowledge, this has been characterized on snow in Antarctica, but not on glacier surface ice, as is stated in the introduction. In the revised manuscript, we will state this more clearly.

Line 49: Define DOC.

We will define DOC on first mention in the revised manuscript.

Line 57: The use of the word "compounds". Were compounds directly measured?
Thank you for spotting this - this should be aliphatic formulae, which will be corrected in the revised manuscript.

Line 61: Please provide definitions of allochthonous and autochthonous DOM pertaining to ice sheets.
We will specify this in the revised manuscript.

Methods:
Line 82: Add a comma after "collection"
Thank you for spotting this - we will correct this in the revised manuscript.

Line 94: Add the word "concentration" after DOC
Thank you for spotting this - we will correct this in the revised manuscript.

Line 95: What is PC?
Polycarbonate - this should have been defined at first mention and will be corrected in the revised manuscript.

Line 96: What is "back in the home laboratory"?
This will be updated to the specific laboratory (GFZ Potsdam).

Lines 83-96: Provide more information about the site names/sample names that correspond with the figure. See next comment about the confusion of the figure and caption for examples. Samples for FTICRMS were only collected at D and Q?
This will be clarified in the revised manuscript.

Figure 1: The lettering in panel B shows up as purple online and is very difficult to discern from the background. When the page is printed in black and white, it's nearly impossible to read. Please change the color scheme to improve clarity. Is site Q on the other side of the stream? Is Q the stream itself? What is SI? The lettering scheme of panel B combined with the lettering scheme of the figure is confusing. My recommendation is to keep the panel lettering scheme and change the panel B lettering scheme of the sites to lowercase letters, numbers, or numerals. The letter "X" is shown in the legend for the sample site, but not used in the figure. What's the difference between field site and sample site?
We will clarify Figure 1 and improve the color scheme in the revised manuscript.

Line 107: Were these odd numbered hours? Please state that. Odd hours is not clear. How many time points were there between 7:00 and 21:00? n=?
Thank you for spotting this! We will clarify this in the revised manuscript.

Line 111: A full description of what?
This was meant to refer to the hydraulic conductivity and water table calculations. This paragraph will be edited for clarity and in accordance with comments from reviewer #1 in the revised manuscript.

Line 125: Please add the word "concentration" after "DOC"
This will be corrected in the revised manuscript.

Line 138 and throughout the text: Please check/correct nitrogen and carbon and N and C for consistency.
This will be corrected in the revised manuscript.

Line 141: What is "back in the home laboratory"?
This will be updated to the specific laboratory (GFZ Potsdam).

Line 151: What is "Predator data station"? If it is a software system, please provide that information.
The methods will be edited and clarified in the revised manuscript to address this and earlier comments regarding details in the methods.

Line 165: Good use of the word "compositions". Check the manuscript to reduce confusing when fluctuating among uses of "compounds", "composition", and "molecular formulae".
This will be checked for consistency in the revised manuscript.

Lines 171-173: This seems important, especially when the use of the term "biolabile" is used often and reported on in the abstract. Was this metric calculated? In Line 179, it states that formulae classifications are informative for this study, but not all mentioned were reported on. Please clarify.
Yes, we used this metric to assess DOM composition but will make this more explicit throughout the revised manuscript for clarity. We will ensure that all mentioned metrics will be reported on in the revised manuscript.

Line 173: Delete the word "above".
This will be corrected in the revised manuscript.

Line 174: Either use "Notably," or start the sentence with "These boundaries…". Delete "It has to be noted that"
This will be corrected in the revised manuscript.

Results:
Figure 2: The purple font is difficult to read in the inset. Is the black triangle the north direction indicator? Is there a significance of the symbol used for Site D in the main figure? That box with the criss-cross in it? Can that be a circular data point instead? Consider using a black outline for

the yellow starting point arrows. They cannot be read in the figure, in the caption, online, and in a printed copy. What is the elevation difference between Site D and the supraglacial stream?

This figure will be clarified with a) a color change, b) removal of the superfluous north arrow, and c) clarification about the location of hole D. The elevation difference is not included as it is not relevant (see response to R1 regarding this figure) but is less than the length of the transit pathway (see scale bar).

Line 200: Add a comma after "TC"
This will be corrected in the revised manuscript.

Lines 200-205: Please report on the results from the leachate DOC concentrations.
We did not report these as the concentrations do not represent anything meaningful in an environmental context. However, for completion we will report them at minimum in the supplementary information and in the revised manuscript we will address this consideration.

Line 207: Provide a definition of what unique molecular formulae are and/or include it as assessment criteria in the methods section.
This will be updated in the revised manuscript.

Lines 207-209: What does "no significant difference between sample types" mean? They all had the same what? And this suggests they were all the same in terms of number of formula assigned? What does that mean? How is it relevant? You can absolutely have the same number of formulae in a bundle of samples from the sample place or in different locations but completely different chemistry/chemistries. Please explain.
We agree that samples can have the sample number of formulae but very different compositions and did not intend to suggest otherwise in this paragraph. We will rephrase this in the revised manuscript.

Line 215: The use of the word "compounds" is confusing. Did you measure compounds or composition, etc. (see major revision comment)? This also shows up in Lines 227. 230, 239, 258, 260, 261, 264, 323, 325, 328, and 382.
This should be formulae / composition throughout and will be corrected in the revised manuscript.

Line 231: This is a good example of the limitations of composition analysis but seems like a random point to make here. Why is this stated here? Discussion section instead? Make the ties to introduction sections that point to microbial and algal blooms?
This section will be rephrased or moved to the discussion in the revised manuscript - it was included in the results section to avoid the presentation of new data in the discussion without presenting them in the results section.

Line 234: Same point for aglycone degradation product as the comment in line 231.
See response to the previous comment.

Line 236: What does "Aromaticity in dark ice was high" mean? Those values look low for aromatic nature (thresholds of AI are 0.5 and 0.67) but are greater than what was reported for the supraglacial stream. Please clarify and consider using more specific terms like "greater" instead of "high". The word "high" is confusing. If it is helpful, put these thresholds in the methods section or further define them here.

This will be corrected in the revised manuscript.

Line 239: Molecular diversity is not the same as number of formulae. Please clarify.

We will use number of formulae throughout in the revised manuscript.

Table 1: Add "(DOC)" after "carbon" and consider adding the identifier "determined by FTICRMS analysis" after "composition" in Line 247. Based on the text, it seems like the biolabile information is missing from the table. Are these results all from Sites D and Q? Define all acronyms in the Table caption, not just RA and #, or point to the methods section for that information.

This will be corrected in the revised manuscript.

Line 267: Add a comma after the word "aromatic"

This will be corrected in the revised manuscript.

Line 266: What's the definition of "significantly different" DOM compositions? Is this based on calculations or chemical characterization?

This was intended to refer to the significant differences reported in Table 1 and the clustering observed in Figure 3A, but we realize that the statement is ambiguous and should be rephrased in the revised manuscript.

Figure 3: Define "PC" in the caption. Point to the methods section for the acronyms and their definitions or provide them here. Are all these data from Sites D and Q?

This will be corrected in the revised manuscript. The data presented here were from sites D, Q and SI - this will be made clearer in the revised manuscript.

Figure 4: Is panel A really all molecular formulae assigned in all samples in the dataset? This looks wrong. Consider moving the word "aliphatic" in panel A to a different location, maybe near the dotted line at H/C = 1.5 on the right-hand side? The chemical character information "aliphatic, HUP, etc." should be included in the caption. Typo for panel E in the caption, change that symbol, which likely was an automatic revision, to (E). The note about the two CHONS formulae may confuse readers that these might be outliers? Are they? Were they a part of a homologous series?

Panel A is the molecular formulae that were assigned in all samples – i.e. only molecular formulae that were assigned in every sample in the dataset are included in this plot, not all molecular formulae assigned across the dataset. The figure will be updated with the suggested edits in the revised manuscript.

Line 287: Be cautious here. The molecular backbone of DOM is CHO containing formulae, not CH, especially since this work doesn't consider hydrocarbon molecular formulae. Classifying CHO as a heteroatom group is incorrect and confusing. Correct the text in the methods section to match.
This will be corrected in the revised manuscript.

Line 293: Molecular diversity is not exclusive to number of formulae. Please correct.
We will use number of formulae throughout in the revised manuscript.

Discussion:
Lines 303-305: This sentence is repeated from the methods. Is it needed here? Please include any discussion of the comparisons across samples.
This will be corrected in the revised manuscript.

Line 308: Edit "DOC" to "DOM"
This will be corrected in the revised manuscript.

Line 310: Edit "concentrations or DOM composition" to "concentrations and/or DOM composition."
This will be corrected in the revised manuscript.

Line 329-330: What does "preferentially leached" mean?
With 'preferentially' here we mean DOM that has already leached from the surface debris and that will therefore not leach from the material again when preparing the water extract. We will rephrase this for clarity in the revised manuscript.

Line 341: Is this statement saying that one molecular formula accounts for 5.7-8.8 %RA of the dark ice DOM samples? How?
Correct – this will be elaborated on to clarify in the revised manuscript.

Line 360: Consider revising "…DOM pools, are hydrologically connected" to "DOM pools, may be hydrologically connected." Measurements of hydrologic connectivity provide results to support what is written, but it is only speculated that that information may come from molecular composition information.
Great point - we will revise accordingly.

Lines 380-382: Confusing and contradictory to what was stated in the previous sentences. What is the main message here? What is the definition of upstream contributions? Just DOM? Debris? Microbes? Processes?
Here 'upstream contributions' was intended to refer to snow melt or surface melt generated at locations above the sampling site. We will rephrase to make this clearer in the revised manuscript.

Line 382: Typo. Delete the extra period, ".", after the parenthesis after "…0.05 %RA)

This will be corrected in the revised manuscript.

Conclusions:
Line 390: The part about microbial communities is confusing/incorrect as there were no biology measurements on microbial communities. Please revise.
This will be corrected in the revised manuscript.